# The Scenario Construction and Evolution Method of Casualties in Liquid Ammonia Leakage Based on Bayesian Network

**DOI:** 10.3390/ijerph192416713

**Published:** 2022-12-13

**Authors:** Pengxia Zhao, Tie Li, Biao Wang, Ming Li, Yu Wang, Xiahui Guo, Yue Yu

**Affiliations:** 1School of Civil and Resource Engineering, University of Science and Technology Beijing, Beijing 100083, China; 2Institute of Urban Safety and Environmental Science, Beijing Academy of Science and Technology (Beijing Municipal Institute of Labour Protection), Beijing 100077, China; 3School of Emergency Management and Safety Engineering, China University of Mining and Technology-Beijing, Beijing 100083, China; 4Safety Culture Research Center, Beijing Academy of Emergency Management Science and Technology, Beijing 100052, China; 5Institute of Smart Ageing, Beijing Academy of Science and Technology, Beijing 100000, China

**Keywords:** liquid ammonia leaked, scenario construction, Bayesian network, emergency management, personnel protection

## Abstract

In China, food-freezing plants that use liquid ammonia, which were established in the suburbs in the 1970s, are being surrounded by urban built-up areas as urbanization progresses. These plants lead to extremely serious casualties in the event of a liquid ammonia leakage. The purpose of this thesis was to explore the key factors of personnel protection failure through the scenario evolution analysis of liquid ammonia leakage. The chain of emergencies and their secondary events were used to portray the evolutionary process of a full scenario of casualties caused by liquid ammonia leakage from three dimensions: disaster, disaster-bearing bodies, and emergency management. A Bayesian network model of liquid ammonia leakage casualties based on the scenario chain was constructed, and key nodes in the network were derived by examining the sensitivity of risk factors. Then, this model was applied to a food-freezing plant in Beijing. The results showed that inadequate risk identification capability is a key node in accident prevention; the level of emergency preparedness is closely related to the degree of casualties; the emergency disposal by collaborative onsite and offsite is the key to avoiding mass casualties. A basis for emergency response to the integration of personnel protection is provided.

## 1. Introduction

In the early 1970s, the food freezing plants have been established by the government on the outskirts of China’s mega cities to ensure that the food supply can meet the rapidly growing standard of living. As the major freezing medium for food-freezing plants, liquid ammonia has the advantages of being green, having a low price, having easy accessibility, and having excellent thermodynamic properties. However, the physicochemical properties of liquid ammonia, such as toxicity, flammability, and corrosiveness, also entail the accident risks of liquid ammonia leakage and fire explosions. With the continuous promotion of the urbanization process and more and more cold storage in the urban built-up areas, surrounding the distribution of residential areas, kindergartens, subway stations, and other densely populated places, the possibility of food freezing plants on the surrounding personnel hazards is gradually expanding.

In order to understand the diffusion process of liquid ammonia leakage and to develop appropriate preventive measures and emergency plans, several studies were conducted to investigate the evolution of accidents based on the unique physical and chemical properties of liquid ammonia. In liquid ammonia leakage experiments, large-scale field experiments are popular; however, due to the high cost of the experiments and the difficulty of reproduction, the experiments are gradually shifting to small-scale experiments, and the research focus is shifting to the numerical simulation of ammonia leakage and dispersion patterns [1]. With the development of computer technology and fluid dynamics theory, vast studies of liquid ammonia leakage simulations based on computational fluid dynamics (CFD) have been carried out, and mathematical simulations of the area that may be affected by liquid ammonia leakage have been conducted [2,3]. In addition, the multi-party study of the toxic gas leak studied the best evacuation path and personal emergency plan for the accident scenario, starting with determining the path and extent of the toxic gas spread [4,5,6]. In summary, it was shown that previous studies on liquid ammonia leaks focused on numerical simulations to determine the physical process of liquid ammonia leakage and diffusion, while the response to liquid ammonia leaks mainly focused on the evacuation path selection, evacuation information dissemination, and evacuation risk evaluation. Existing studies failed to reflect the evolution of liquid ammonia leakage rapidly from the whole space and time inside and outside the built-up urban areas, and it was also difficult to find the key nodes in the evolutionary process to stop the spread of the accident; thus, targeted measures cannot be effectively taken to prevent and control the key nodes that lead to liquid ammonia leaks, and the emergency rescue after an accident may also be difficult to carry out effectively due to the lack of relationship between the source of the leak and other scenes in the vicinity. Therefore, from the perspective of the relationship between the source of liquid ammonia leakage and other surrounding scenes, this paper explores the evolution of the whole scene in the leakage place and the surrounding places after the liquid ammonia leakage accident and then provides the basis for the joint prevention and control inside and outside the liquid ammonia leakage place.

In this paper, we took liquid ammonia as a causative factor from a global perspective and constructed a full scenario evolution process for liquid ammonia leakage by using the scenario construction method. The word scenario first refers to a description of a set of facts or states that can move a state of affairs from an initial state to a future state [7]. Different scholars have different opinions and descriptions [8,9]. Scenario construction can help decision-makers find the set of events and elements that cause the target scenario system to move from the current state to future development and, thus, provide contingency plans to hinder an undesirable evolution. At present, scenario building is widely used in scenic spots, railroads, cities, mines, and other scenarios, aiming to summarize and extract key elements of various emergencies in different scenarios and then carry out emergency capability assessment, scenario evolution analysis, and association rule mining, with response strategies and accident evolution probability calculation in the respective application scope [10,11,12,13]. In addition, in the field of material leakage and diffusion research, including marine oil spills and chemical tank leaks, scenario-based construction provides a strong basis for managers to develop prevention and control plans, implement emergency decisions, and analyze the causes of accidents [14,15].

In addition, to explore the key nodes leading to liquid ammonia leakage, this paper transforms the full scenario evolutionary flow chart obtained from scenario construction into a Bayesian network through the bow-tie model. As a model based on probabilistic analysis and graph theory, a Bayesian network is designed to collect and represent knowledge in uncertain domains and can perform probabilistic calculus and statistical analysis efficiently. It shows excellent applicability in applications targeting risk analysis and reliability, such as gas pipeline accidents, dust explosions, dam failures, and tsunamis [16,17,18,19]. It also developed various damage risk assessment models and human reliability analysis methods, which greatly enhanced the connection and development between various disciplines [20,21]. In addition, it also showed good applicability in the process of scenario evolution. By combining Bayesian networks with scenario evolution analysis methods in different scenarios, scholars have established the evolution probabilities and consequences of various events in different scenarios and made corresponding contributions to emergency management [13,22]. At the same time, in the modeling process, a Bayesian network is not only easy to use, but it also can obtain a model that can clearly express the logical relationship between the nodes and has more accurate results.

In order to explore the key control nodes of personnel protection failure in the process of liquid ammonia leakage in food-freezing plants in urban built-up areas, based on the theory of public safety triangle, this paper innovatively combined the scenario construction method, a Bayesian network, and the bow-tie model to construct a Bayesian model of casualties caused by liquid ammonia leakage. The full paper constructed and improved the full scenario evolution process of liquid ammonia leakage from the perspective of the full scenario evolution of liquid ammonia leakage, and by examining the sensitivity of risk factors, the key nodes of accident prevention and personnel protection were obtained, which fills the research gap in the field of full scenario evolution of liquid ammonia leakage and provides strong support for managers of food-freezing plants in megacities and surrounding areas and relevant government departments to formulate liquid ammonia leakage response plans.

## 2. Method

Based on the public safety triangle theory, this study clarified the relationship between the liquid ammonia leakage source and other surrounding scenarios through scenario construction, and formed a perfect scenario evolution path. In order to realize the key node control in the evolution path, the article took the bow-tie model as the mediator, transformed the constructed full scenario evolution process into a Bayesian network, and realized the identification of key nodes through inverse reasoning and sensitivity analysis. In order to maximize the accuracy of the results obtained, the article used the survey method, Dempster-Shafer evidence method, and empirical research method to analyze the data obtained through the questionnaire to ensure the objectivity of the data as much as possible and further ensure the accuracy of the results.

### 2.1. Scenario Construction of Liquid Ammonia Leakage

The process of scenario response could be divided into scenario concept, scenario description and element extraction, scenario evolution, scenario network construction, and scenario projection [12]. Due to the dynamic and unpredictability of emergencies, how to use existing situations to infer the development of future emergencies trend is the focus at present. Due to the characteristics of secondary and derivative emergencies transforming into each other and highly interrelated emergency management in the process of liquid ammonia leakage, and, based on the public safety triangle theory, this paper constructed the evolutionary structure of the human casualty scenario chain after liquid ammonia leakage by studying accident cases [23].

The public safety triangle theory divides the public safety system into three parts: emergencies, carriers, and emergency management. According to the public safety triangle theory, liquid ammonia is identified as a disaster factor. If the human intervention measures that can prevent or reduce the emergencies fail, once the liquid ammonia exceeds the critical amount or encounters some trigger conditions (such as pipeline corrosion, aging, personnel misuse, etc.), a leak may occur, resulting in a liquid ammonia leakage emergency. At this moment, the disaster elements carried by the emergency event will act on the carrier, changing the state of the carrier in the form of material and other manifestations and causing the destruction of the carrier itself or the loss of some functions. The emergency management of the carrier is an important means of prevention and control to reduce accident casualties, which is of great significance to reduce the consequences of accidents.

Six types of bearing carriers were identified via summarizing, analyzing, and concluding liquid ammonia leakage accidents, which are refrigerating plants, residential areas, subway stations, kindergartens, universities, and shopping malls. As an example, the scenario construction of refrigeration plants is shown in Figure 1. Figure 1 shows that, as a disaster element, the process of liquid ammonia leakage when it is in a storage tank or outlet valve pipeline is susceptible to a variety of factors, such as the corrosive effect of liquid ammonia on the pipeline, natural disasters or third-party damage, and aging failure of storage tank equipment. When emergency management measures for this carrier fail, a liquid ammonia accident will occur. Once an accident occurs, the workers will bear the brunt of the liquid ammonia damage. Due to the liquid ammonia leakage with gas-liquid two-phase diffusion, it is easy to cause further expansion of the accident if the emergency response capability for liquid ammonia leakage is insufficient and the monitoring is not standard. Finally, the accident will act on other carriers, poisoning the people around the refrigeration plant. Through the analysis and summary of the corresponding bearing carriers, a full scenario evolutionary flow chart of liquid ammonia leakage was derived, as shown in Figure 2.

### 2.2. Scenario Construction Based on Bayesian Network

#### 2.2.1. Bayesian Network Model

As a combination of directed acyclic graphs (DAG) and probability theory, a Bayesian network (BN) mainly consists of nodes and directed edges that reflect information about the target problems and describe the relationships between different nodes [24]. Nodes in the network can be divided into two types of nodes: parent and child. Among them, if there is a directed arc from X_j_ pointing to X_i_, then X_j_ is called the parent node of X_i_ (denoted as Pa(X_i_)), and X_i_ is the child node of X_j_ (denoted as Ch(X_i_)). A node in a Bayesian network without parent nodes is called a root node, and a node without a child node is called a leaf node. The parameters in the network include marginal probability distribution (MPD), to which the root node is attached, and conditional probability distribution (CPD), to which the non-root node is attached. Based on the d-separation criterion, Bayesian network can determine the conditional independence relationship between the nodes quickly [25]. For a node variable X_i_ in the network, Pa(X_i_) is the set of variables on which the node directly depends, and X_i_ is conditionally independent of variables that do not belong to its parent node Pa(X_i_) and can be written as:P(X_i_|X_1_, …, X_i−1_, X_i+1_, …, X_n_) = P(X_i_|Pa(X_i_))(1)

With the help of this feature, the Bayesian network is able to decompose joint probability distributions into products of simple conditional probability distributions, as follows:(2)$$P(X_{1},\;X_{2},\ldots ,\;X_{n})=\prod_{i=1}^{n}P(X_{i}|Pa(X_{i}))$$

Figure 3 illustrates a simple Bayesian network. A is the root node, E is the leaf node, B is the child node of A, C and D are the children of B, and A, B, C, and D are the ancestor nodes of E. In this sample, the probabilities of each node can be calculated by MPD and CPD. Assuming there are only two discrete states of the nodes in the network, the joint probability distribution under this Bayesian network framework is as follows:P(A, B, C, D, E) = P(A) P(B|A) P(C|B) P(D|B) P(E|C,D)(3)

The posterior probability of a child node in a Bayesian network is influenced by its parent node and varies with the prior probability of the parent node or new evidence. This gives Bayesian networks two important functions: reverse deduction and sensitivity analysis. The reverse deduction function means that if a state occurs with 100% for a node, the probability of its parent node also changes, and the higher the change in the probability of the parent node, the greater the impact will have on that child node.

There are three main steps in constructing a Bayesian network. The first one is identifying the Bayesian variables and their state classifications. The second is establishing the structure to determine the causal relationships among the variables in the network. The final step is determining the conditional probabilities of all nodes [13]. In this paper, based on the scenario construction evolutionary flowchart, the Bayesian network model was built in conjunction with the bow-tie model.

#### 2.2.2. BN Variables of Liquid Ammonia Leakage

Through scenario construction for liquid ammonia leakage emergencies, a whole chain of liquid ammonia leakage scenarios can be established, and six major carriers as well as the consequences of accidents can be identified. Scenario-building technology is the application of “bottom-line thinking” in the field of emergency management, which is essentially a process of risk analysis and hazard identification [11]. However, to obtain further information on the causes and consequences of the accident, an additional approach is needed for in-depth analysis.

The bow-tie model is one of the best graphical ways to represent accident scenarios, as it can clearly describe the cause and results of accidents. It is an accident causal analysis method that integrates fault tree and event tree. The bow-tie model is now widely used in risk analysis due to its ability to integrate the causes and consequences of accidents into a single graphical model [26,27]. However, the bow-tie model also has limitations and is not suitable for systems with redundant failures, multiple state variables, and correlated events, and it is difficult to handle uncertainty in risk analysis unless combined with other techniques [27,28]. Bayesian networks are now widely used in the security risk analysis of dynamic systems and are used to overcome the limitations of conditional dependencies between events in bow-tie analysis [19,29]. Therefore, this paper combined a bow-tie model and a Bayesian network to determine the key causes and possible consequences of liquid ammonia leakage.

Through accident trees and event trees, various risk factors were further transformed into Bayesian networks based on the six scenarios identified by scenario construction. The conversion process is shown in Figure 4 [30].

Based on the bow-tie model transformation rule, when building a Bayesian network, it is necessary to determine the fault tree generated by the liquid ammonia leakage accident firstly. As a common assessment method in risk assessment, a fault tree is used to identify and evaluate the hazards of various systems using logical reasoning, which can not only analyze the direct causes, but also reveal the potential causes of accidents in depth to find out the inner law between factors, thus establishing the basis for determining the causal relationship of Bayesian network nodes [31]. Therefore, the fault tree of a liquid ammonia leakage accident was drawn as shown in Figure 5. Appendix A provides a detailed supplement to Figure 5.

After that, the event tree of the liquid ammonia leakage acting on the other scenarios was drawn. An event tree is an important method that can visually and graphically show the various combinations of event sequences and accident consequences likely to occur after the initial event. In the analysis process, liquid ammonia leakage was considered the initial event, and then the adverse consequences of the accident expansion caused by the liquid ammonia leakage acting on the refrigeration plant were analyzed. The results of the scenario construction showed that, if the liquid ammonia leakage is not prevented in time, then the accident will successively act on the surrounding buildings after further expansion. Taking the residential area as an example, an event tree was created as shown in Figure 6.

According to the scenario construction diagram, it was determined that the liquid ammonia leakage accident will cause poisoning and other consequences after acting on the refrigeration plant, and if the liquid ammonia leakage accident continues to expand at one time, it will successively act on the other five scenarios. Thus, the event tree of liquid ammonia leakage accidents acting on other scenarios were drawn, respectively, and then a Bayesian network was constructed combined with the bow-tie model transformation rule, as shown in Figure 7. The name and information of nodes are shown in Appendix A.

## 3. Application

### 3.1. Applicable Analysis

#### 3.1.1. Simulation Scenario

Taking a food-freezing plant in Beijing as an example, the established liquid ammonia leakage scenario evolution process and Bayesian network model were used to analyze the freezing plant. The factory uses liquid ammonia as refrigerant in production, and its maximum use of liquid ammonia is 15 t, which is greater than the specified critical amount of 10 t (GB18218-2018, 2018) [32], constituting a Class III major hazardous source of hazardous chemicals, being one of the local major hazardous sources of liquid ammonia. At the same time, the factory is surrounded by flat landscapes and crowded buildings, and in the surrounding area, there are kindergartens, residential buildings, and other dense places, as shown in Figure 8a.

From the analysis of physical factors, due to the impact of temperature, the food refrigeration industry significantly enhances the cooling capacity in the summer, resulting in the increased work intensity of machinery and equipment, further increasing the probability of equipment failure. From the human factor viewpoint, staffs are more likely to be fatigued in the summer, resulting in lower attention and judgment. Therefore, the accident scenario was established at noon on a Friday in mid-August. By looking up the meteorological data (http://tianqi.2345.com, accessed on 16 August 2021), it was found that the daytime temperature in the area was from 30 °C to 36 °C in August, with the wind below level 3, mostly southwest or no sustained wind, thus setting the daytime temperature on the day when the accident occurred as 32–35 °C, with 3.5 m/s wind and southwest directions.

Based on that, the Norwegian DNV’s SAFETI software was used to simulate the impact of the liquid ammonia leakage in the factory, providing objective evidence for the Bayesian network analysis. There were two modules, consisting of a quantitative accident consequences calculation module and a risk calculation module, which can be used to calculate the flow rate and state of material leakage into the atmosphere. Since liquid ammonia leakage is easily affected by working temperature, tank pressure, and tank volume, and through research and finding governance, the simulation parameters and other information were arranged as shown in Table 1.

#### 3.1.2. Simulation Result

In the simulation software, the AEGL indicator (Acute Exposure Guideline Level) was applied to describe the toxic damage area of the liquid ammonia leakage. AEGL-1 indicates that concentrations of air toxins above this level produce significant discomfort, irritation, or certain asymptomatic non-sensory effects in the general population, including susceptible individuals. However, these effects are not disabling and are transient, being reversible when exposure ceases. AEGL-2 indicates that, at concentrations of air toxins above this level, the general population, including susceptible individuals, will experience irreversible or other serious, prolonged adverse health effects or diminished ability to escape. AEGL-3 indicates that concentrations of air toxins above this level result in life-threatening health effects or death in the general population, including susceptible individuals.

The simulation results of the liquid ammonia leakage are as shown in Figure 8b. Figure 8b illustrates that, at 3.5 m/s wind speed, once a liquid ammonia leakage occurs, the farthest coverage area of liquid ammonia toxic lethal area is 70 m (AEGL-3, exposure concentration above 1100 ppm in 60 min), the farthest coverage area of liquid ammonia toxic serious injury area is 276 m (AEGL-2, exposure concentration above 160 ppm in 60 min), and the farthest coverage area of liquid ammonia toxic minor injury area is 715 m (AEGL-1, exposure concentration above 30 ppm in 60 min). Then, MARPLOT software was applied to import the simulation results into Figure 8a, obtaining the site coverage area of liquid ammonia leakage, as shown in Figure 8c.

Figure 8c indicates that, when the simulated wind direction is southwest, the red area is the liquid ammonia toxic lethal area (AEGL-3), the orange area is the liquid ammonia toxic serious injury area (AEGL-2), and the yellow area is the liquid ammonia toxic area minor injury area (AEGL-1), where the yellow boundary is the region that may be covered by liquid ammonia under the actual wind direction. In summary, once a liquid ammonia leakage occurs, if unstopped, the impact can extend far beyond the plant boundaries and threaten people in the surrounding establishments.

### 3.2. Determination of Conditional Probability

For further scenario extrapolation and to determine the probability of casualties, the probability of each type of node needs to be determined. For the root nodes in the Bayesian network, the prior probabilities of some of the root nodes were determined by 73 accident cases from 2010–2022, which were obtained from the emergency management websites of the provinces and cities. Such root nodes mainly include X4, X8, X11, etc. The specific prior probability was mainly calculated by frequency; for example, among the 73 accident cases collected, 24 accidents were caused by pipe rupture, and the occurrence probability was 0.33. In addition, for some of the root nodes that were difficult to translate through accident cases, such as the risk identification ability of personnel, we adopted the approach of Wu et al. and conducted a questionnaire survey [13]. The survey aimed to investigate the experts’ estimation of the probability of the occurrence of the node, and the priori probability was given directly from the experts’ experience and knowledge; it included questions such as “What do you think is the probability of the occurrence of an accident in which the workers’ risk recognition ability is not strong?”. Subsequently, the results were processed using the Dempster-Shafer evidence method and the prior probabilities of such nodes were given. Finally, the conditional probability tables of the sub-nodes were determined according to the structural form of the bow-tie model and the opinions of the experts.

An example of a liquid ammonia leakage to a college was used here to determine the node probability. We investigated the opinions of four experts and obtained the initial prior probability values. Due to the subjectivity and uncertainty of the experts’ opinions, we adopted the Dempster-Shafer method to process the collected data, taking into account the opinions of each expert. The Dempster-Shafer evidence method is shown in Figure 9, and the conditional probability values of the nodes were calculated by this method. The results are shown in Table 2.

### 3.3. Results

#### 3.3.1. Results of Empirical Analysis

By determining the prior probabilities of each parent node and the conditional probabilities of the child nodes, a Bayesian network model was built using the software GeNle4.0, and the probability values of each node were entered into the software for subsequent empirical analysis.

Through the Bayesian network model, the probability of each node in the whole network was calculated, resulting in an intuitive grasp of the overall dynamics of the liquid ammonia leakage accident and accident expansion. In the past few years, liquid ammonia leakages showed various characteristics; thus, it is necessary to find the key nodes that caused the accident through the reverse reasoning of a Bayesian network. Reverse reasoning is the most commonly used method in Bayesian networks, which means that, by predetermining the probability of a liquid ammonia leakage occurring and reasoning about the probability of each causal factor under that probability, the most likely cause of the liquid ammonia leakage can thereby be diagnosed [33]. It is possible to calculate the posterior probability of each factor by reverse reasoning, and the larger the posterior probability, the greater the impact on the outcome event. Setting the liquid ammonia leakage as “occurrence” (100%), the posterior probability of each node was obtained, and its comparison with the change of the prior probability is shown in Figure 10. Figure 10 shows that the posterior probabilities of the four causal factors X3, X11, X4, and X2 were higher when the liquid ammonia leakage occurs. In other words, when liquid ammonia leakage accidents occur, inadequate risk identification ability (X3), pipeline break (X11), insufficient professional skills (X4), and loose safety management (X2) are more likely to occur. Among them, the factor “X11” had the highest prior probability with a value of 33%, which indicates that the key factor of a liquid ammonia leakage in previous experience and analysis was pipeline break. Among the human factors, risk identification capability had the highest posterior probability, with a value of 40.9%, which indicates that inadequate risk identification capability is the most likely key risk factor for accidents. This situation shows the failure to detect abnormalities during production or maintenance due to inadequate personnel risk identification capabilities, resulting in situations such as pipeline breaks and causing liquid ammonia leakage accidents.

Identifying key causative factors only by prior or posterior probabilities may lead to inaccurate results; thus, sensitivity analysis was used to validate the results [34]. Sensitivity analysis is achieved by studying the effect of small changes in the numerical parameters of the model (prior and conditional probabilities) on the output parameters (posterior probabilities). Using GeNIe, a complete set of derivatives of the posterior probability distribution of the target node over each numerical parameter of the Bayesian network was efficiently computed, and these derivatives showed the importance of the numerical parameters in the network for computing the posterior probability. If the derivative of a parameter is large, then a small change in that parameter may result in a large change in the posterior probability of the target node; conversely, a derivative that is too small has little effect on the posterior probability. Therefore, by setting the liquid ammonia leakage event T as the target node and using the sensitivity analysis function of the software, the analysis results as in Figure 11 and Figure 12 were obtained. In the sensitivity analysis, the redder the color of the root node in the network, the higher the sensitivity; as shown in Figure 11, X1, X2, X3, X4, X5, X17, and X18 have the most striking color in the whole network. To further obtain the key root nodes in the network, we plotted a tornado diagram for sensitivity analysis, which shows the parameters most sensitive to the selected states of the target nodes, ranked from most sensitive to least sensitive. Figure 12 shows the range of the target state when the parameter was varied within its range as [−10%, 10%]. The values of the derivatives and some coefficients of the root nodes in Figure 12 are included in Table 3, and it can be seen that node X3 had the highest derivative value D. This indicates that a small change in the parameters of node X3 produced a huge change in the posterior probability of the target node, which means that X3 may be the key node of the network. Therefore, when controlling liquid ammonia leakage, training and education for employees should be strengthened, focusing on the risk identification ability of personnel. In addition, other nodes with high sensitivity also need to take timely measures for risk prevention.

#### 3.3.2. Results of Emergency Preparedness Analysis

Due to the physical and chemical properties of liquid ammonia, once a liquid ammonia leakage occurs, if corresponding measures are not taken in time, then the liquid ammonia leakage will expand rapidly. In order to predict the possible expansion of the liquid ammonia leakage, “Forecast Alarm” was set to “no (100%)”, “Automatic Protection System” to “no (100%)”, and “Close Valve and Plug Leak” was only set to “yes (100%)”, and the range of liquid ammonia leakage at this time is shown in Table 4. It was found that the probability value of liquid ammonia leakage in the range of 200–500 m was the largest at 42.1% when all safety protective measures failed.

Due to the influence of distance and various factors, in the case of different expansion ranges of liquid ammonia leakage, the casualties faced by various carriers were also different. In this paper, the probability of casualties caused by liquid ammonia leakage to other scenarios under different leakage range scenarios was also recorded separately, as shown in Table 5. As can be seen from Table 5, in the case of different liquid ammonia leakage ranges, when the probability of other nodes was constant, the probability of casualties in kindergartens was generally the highest, reaching 32.9%, while the probability of casualties in colleges was generally the lowest.

Emergency preparedness is an important factor in preventing accidents from occurring and expanding, and the importance of emergency preparedness needs to be emphasized when it is determined that there is a high probability that a liquid ammonia leakage could spread to various scenarios. Since the most likely area for liquid ammonia leakage to spread is concentrated within 200~500 m of the liquid ammonia leakage site, and the kindergarten happened to be closer to the liquid ammonia plant, which had the highest probability of casualties, it is necessary to strengthen the multi-body coordinated emergency response resource allocation and preparation for the kindergarten.

Taking kindergartens as an example, considering the characteristics of the young age of kindergarten personnel and their close distance to the freezer plant, it was of great significance to analyze the impact of node status related to kindergarten emergency preparedness on the change of casualties in kindergartens. According to the “National Emergency Response Plan for Environmental Emergencies”, emergency preparedness mainly includes early evacuation and emergency resource allocation [35]. Therefore, the status node related to emergency preparedness was set to “no (100%)”, and the leak to kindergarten was set to “yes (100%)”, without changing other nodes. At this point, the probability of causing casualties in the case of liquid ammonia being able to leak into the kindergarten scenario (yes, 100%) without changing the state of the node related to emergency preparedness was about 36%, while the probability of casualties rose rapidly to 53.2% when the state of inadequate emergency preparedness was set. Since most kindergarten students are younger children, they do not have higher safety awareness compared to adults, and their ability to escape or save themselves in an emergency is very weak; thus, necessary emergency preparedness is of great importance to ensure the safety of kindergarten teachers and students.

Repeating the above, the probability values of casualties of other scenarios under different emergency preparedness states were counted as shown in Table 6. It can be seen from Table 6 that, except for kindergarten when emergency preparedness is insufficient, the probability of casualties in subway stations will also rise rapidly to 68.3%. This is because in the subway, as a city transportation hub, the passenger flow is generally large, and the subway station is often in an underground space and does not have good ventilation conditions. Once the liquid ammonia leakage spreads to the subway station, then it is very likely to cause panic, resulting in emergency evacuation difficulties and other problems, which will lead to the emergence of a large-scale poisoning phenomenon. In addition, in the other three scenarios, there was no more than 25% increase in the probability of casualties, and the probability of casualties was generally low.

#### 3.3.3. Results of Emergency Disposal Analysis

The timeliness of emergency disposal is a key factor to reduce accident casualties. Emergency disposal mainly includes the emergency evacuation taken by each scene after the occurrence of liquid ammonia accidents, the emergency rescue of relevant personnel, and the emergency mobilization and other activities of relevant departments according to the early warning information. In emergency disposal, each departmental unit should mobilize local emergency resources and organize emergency forces to carry out timely and effective emergency disposal activities.

We explored the importance of integrated emergency disposal by comparing the casualty probability caused by emergency disposal in the refrigeration plant only and the integrated emergency disposal in and out of the plant after liquid ammonia leakage accidents. First, the emergency disposal node related to the refrigeration plant was set to yes (100%), and then the emergency disposal node related to each scenario was set to untimely or unsuccessful (0) without changing the rest of the nodes to obtain the casualty probability of each scenario in the emergency disposal state of the plant only. After that, the emergency disposal node of each scenario was set to in time or successful (100%) to obtain the casualty probability of each scenario in the integrated emergency disposal state in and out of the plant, and the results are shown in Figure 13.

First of all, in the case of liquid ammonia leakage, other nodes in the plant remained normal and the nodes related to emergency disposal were adequate. The probability value of liquid ammonia leakage ranging within 200 m was the largest, reaching 85%, which means that, in the case of timely emergency disposal in the refrigeration plant, it is almost difficult to threaten the scenes above 500 m distance, but other scenarios still had the probability of human casualties. Secondly, the probability of producing casualties in kindergartens was the largest when only the emergency disposal of refrigeration plants existed, reaching 29.8%, followed by subway stations, residential areas, and malls, respectively, and the probability of producing casualties in universities was less than 0.1%. Finally, in the state of integrated emergency disposal in and out of the plant, the casualty probability in each scenario decreased rapidly, and the final probability of casualties did not exceed 5%. Among them, the probability of kindergarten casualties fell the most at 28.2%, while the probability of subway station casualties fell the least, except for colleges, for which a probability of casualties was 0.6%. In short, the importance of the integration of emergency disposal inside and outside the plant is self-evident; thus it is necessary to strengthen the integration of emergency disposal between the major scenarios with the refrigeration plant to achieve the coordinated operation of multiple subjects and to effectively reduce the probability of casualties and safeguard people’s lives.

## 4. Discussion

Due to the dynamic and uncertain nature of liquid ammonia leakage, in the actual evolution of the accident, there are both unexpected and expected evolutionary directions, and it is often difficult to control the situation in time, which can easily cause casualties in the plant and surrounding scenes [22]. When the scenario evolution path of a liquid ammonia leakage is consistent with the desired management objectives, the development of the accident can be considered to be effectively controlled. Therefore, emergency decision-makers should be cautious and take reasonable measures to prevent further expansion of the leak and loss of life whenever possible. In order to help decision-makers achieve timely and effective decision-making, this paper adopted a scenario-building approach based on the public safety triangle model, focusing on three dimensions of disaster elements, carriers, and emergency management, combed 73 liquid ammonia leakage accident cases in the past ten years, and established a full-scenario accident evolution chain for liquid ammonia leakage. After that, SAFETI software was used to simulate the spreading range of the accident under uncontrolled conditions and clarified the possible toxic areas of liquid ammonia leakage. Further, the bow-tie model was adopted to analyze the liquid ammonia leakage accident, a full-scene Bayesian network of liquid ammonia leakage was established, and the probability of accident expansion and casualties was clarified by combining historical cases and expert knowledge.

By applying the AEGL index to the simulation analysis, it was found that, under normal circumstances, in the event of an unstopped liquid ammonia leakage, the farthest coverage of the lethal zone, the serious injury zone, and the minor injury zone would reach 70 m, 276 m, and 715 m, respectively, with exposure concentrations of 1100 ppm, 160 ppm, and 30 ppm within 60 min. This means that if a liquid ammonia leak occurs and is left unchecked, the impact of the leak will extend far beyond the plant area and pose a significant threat to people in the surrounding scenes.

To further clarify the antecedents and consequences of liquid ammonia leakage, a bow-tie model was adopted and a Bayesian network was established. In this paper, we used accident cases and expert opinions to clarify the probability of each node and used inverse reasoning to analyze the most likely cause of liquid ammonia leakage. The results showed that X11 (pipe break) had the highest a priori probability, while X3 (insufficient risk identification) had the highest a posteriori probability. To clarify the key nodes of liquid ammonia leakage accidents, sensitivity analysis was adopted for in-depth analysis. The network node colors indicated that nodes X1, X2, X3, X4, X5, X17, and X18 had high sensitivity. In addition, after combining the derivative values of each bar in the tornado diagram, the root node X3 had the strongest sensitivity, which indicates that a small change in the probability of node X3 will cause a large change in the posterior probability of the occurrence of a liquid ammonia leakage at the target node. Combined with the fact that X3 had the highest posterior probability in the inverse reasoning, which means that X3 was the key root node, the lack of personnel risk identification ability was an important cause of the accident; therefore, the refrigeration plant should focus on strengthening the risk identification ability of personnel in the risk prevention process to prevent the occurrence of liquid ammonia leakage accidents from the human factor perspective.

Analysis of the expansion range of liquid ammonia leakage showed that the highest probability of expansion of liquid ammonia leakage is between 200 and 500 m. Among the different distributions, the overall probability of casualties was generally higher for kindergartens and lowest for colleges. The reasons for this result are more related to the distance of different sites from the source of the liquid ammonia leakage and are also influenced by the level of emergency preparedness and emergency response at different sites. In the subsequent analysis, by varying the probability values of the nodes related to emergency preparedness, it was found that the probability of casualties increased in both cases of inadequate emergency preparedness, with the probability of casualties in kindergartens and subway stations largely exceeding 50%. This indicates that the probability of casualties in the case of accidents with more adequate emergency preparedness is smaller, which is consistent with the results of some studies [36,37].

Finally, the synergistic resource allocation and emergency preparedness mechanisms of multiple entities have a significant impact on the probability of human casualties in liquid ammonia leakage accidents in food refrigeration plants. Studies show that, in a liquid ammonia leakage situation, when the refrigeration plant emergency response is timely, it is almost difficult to threaten scenarios beyond a distance of 500 m, but other scenarios may still have casualties. The results of the analysis showed that the probability of producing casualties in the kindergarten in the presence of only emergency disposal of the refrigeration plant reached 29.8%, and in the state of integrated emergency disposal inside and outside the plant, the probability of casualties in the kindergarten dropped significantly to 1.6%. The probability of casualties in each scenario, in that case, did not exceed 5%, the importance of which cannot be overstated. Therefore, it is necessary to accelerate the construction of integrated emergency response capabilities inside and outside the plant to achieve the coordinated operation of multiple entities, thus effectively reducing the probability of casualties and safeguarding people’s lives.

In conclusion, the conclusions drawn from the empirical analysis in this paper can assist decision-makers in grasping the evolutionary path of liquid ammonia leakage accidents and have important practical significance for decision-makers to take measures to prevent accidents from occurring. In addition, by further identifying key nodes, it can help workers achieve the precise control of risks, which plays a great significance to the safety work. Furthermore, this paper organically combined scenario construction, the bow-tie model, and the Bayesian network to empirically analyze the common causal factors and consequences of liquid ammonia leakage accidents, which can precisely locate the probability of accident occurrence and expansion, effectively improve the emergency management of liquid ammonia leakage in food refrigeration plants in urban built-up areas, fill the research gap in the field of liquid ammonia leakage scenario evolution, and provide a theoretical reference to further promote the research of liquid ammonia leakage accident evolution mechanisms.

However, there are still some limitations in this paper. First, it was found in the accident collection that the current liquid ammonia leakage accidents were not systematically counted, making it difficult to obtain detailed information on the occurrence of accidents, forcing most nodes to adopt expert scoring to determine probability values. Although we used D-S evidence theory in the subsequent process to fully consider the opinions of individual experts, the results obtained may still deviate from the actual situation. In addition, traditional Bayesian networks are used for analysis; however, the uncertainty of random events in many cases also varies with time, making it difficult to model and reason about dynamics. In the future, a database of liquid ammonia leakage accidents should be established to calculate the a priori probability and conditional probability of nodes using objective cases. In addition, the application of Bayesian networks in liquid ammonia leakage accidents should be developed in the future to achieve dynamic modeling and inference.

## 5. Conclusions

Based on the public safety triangle model, this paper adopted scenario construction and combined Bayesian networks to estimate and analyze the probability of occurrence and consequences of liquid ammonia leakage accidents in food refrigeration plants in various scenarios. Specifically, this paper selected scenarios such as kindergartens and residential areas as the objects of liquid ammonia leakage, constructed a full-scene evolutionary chain of liquid ammonia leakage and diffusion, and specifically analyzed the causal relationships among the influencing factors by combining the bow-tie model, and it then formed a Bayesian network for the evolution of liquid ammonia leakage accidents. This paper also took a food freezing plant in Beijing as an example, adopted SAFETI software to determine the range of possible effects of liquid ammonia leakage, determined the probability values of various nodes in the network through accident cases and expert knowledge between 2012 and 2022, and conducted an empirical analysis of the consequences of the accident. The results showed that, at a wind speed of 3.5 m/s, once a liquid ammonia leakage occurs, the farthest coverage areas of the liquid ammonia toxic fatality area, serious injury area, and minor injury area are 70 m, 276 m, and 715 m respectively, with a very wide impact range. In addition, through reverse reasoning and sensitivity analysis, it was clear that the key node of liquid ammonia leakage accident was X3 (poor risk identification ability of personnel), which indicates that enhancing the risk identification ability of personnel through training and education is the key to prevent liquid ammonia leakage accidents. Further studies show that once the safety protection measures in a refrigeration plant fail, then the spread of a liquid ammonia leakage increases significantly, with the greatest probability of a liquid ammonia leakage in the 200–500 m range when all measures fail, which causes the surrounding scenes to be at risk of injury or death. Among them, the probability of casualty is generally higher in kindergartens at different leakage ranges. In addition, the results of the study showed that the probability of casualties at subway stations climbs to 68.3% in case of inadequate emergency preparedness, and the probability of casualties in the remaining major scenarios also increases significantly, which highlights the importance of emergency preparedness. Similarly, by comparing the casualty probability of emergency disposal only inside the plant and the integrated emergency disposal inside and outside the plant, the casualty probability of each scenario under the integrated emergency disposal inside and outside the plant decreased rapidly, and the final casualty probability did not exceed 5%, which shows the importance of integrated emergency disposal inside and outside the plant.

In conclusion, this study provides a scientific basis for the development of effective liquid ammonia leakage prevention measures and accident prevention means for food-freezing plants and relevant government departments that are gradually entering built-up areas in the process of urbanization, and it lays the foundation for achieving control of key nodes and effective emergency rescue, thus reducing casualties. In addition, the research method adopted in this paper constructed a full-scenario evolutionary process of liquid ammonia leakage, which, combined with a Bayesian network, rapidly carried out the extrapolation and analysis of a liquid ammonia leakage accident, which provides a reference for future research on liquid ammonia leakage accident prevention and fills a gap in the current literature in the field of research on the action and diffusion process of liquid ammonia leakage accidents. However, there are certain limitations in this paper, such as the subjectivity of the data and the inability to dynamically track the evolution of the accident development process. Future research needs to conduct more research in improving the precise control of the dynamic evolution mechanism of liquid ammonia leakage, improving the objectivity of the probability of each node in the liquid ammonia leakage network, and correcting the evolution process among networks.

## Figures and Tables

**Figure 1 ijerph-19-16713-f001:**
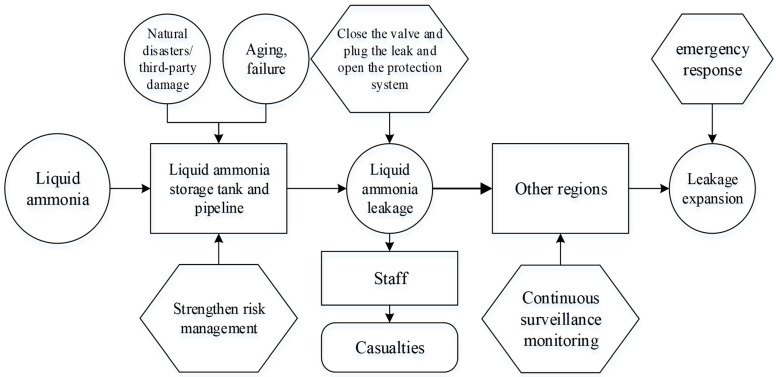
The evolution process of liquid ammonia leakage scenario in the refrigeration plant.

**Figure 2 ijerph-19-16713-f002:**
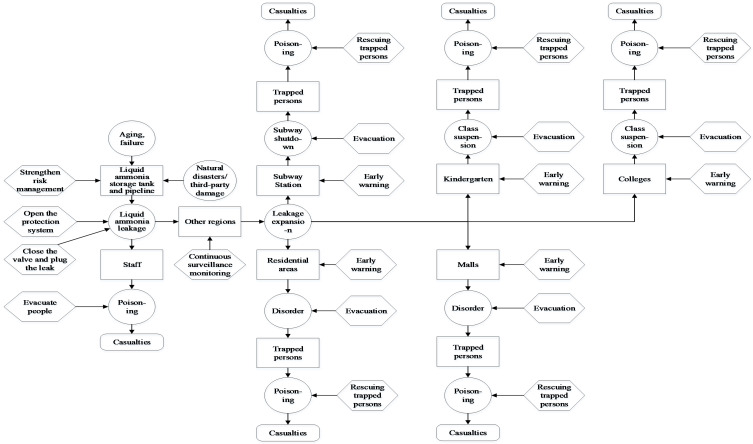
Liquid ammonia leakage full scenario evolution flow chart.

**Figure 3 ijerph-19-16713-f003:**
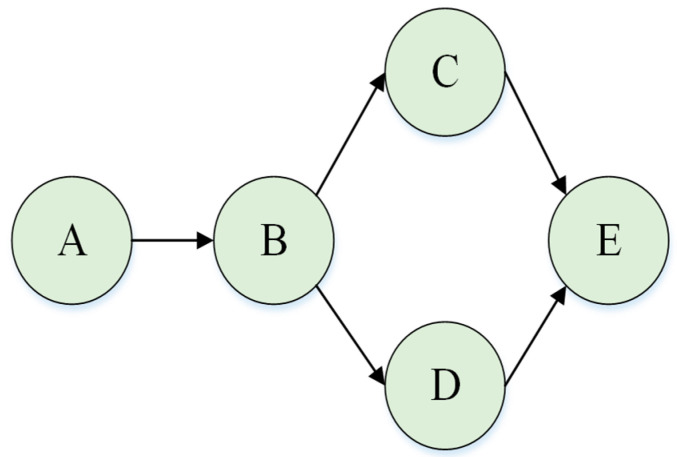
Sample of a Bayesian network.

**Figure 4 ijerph-19-16713-f004:**
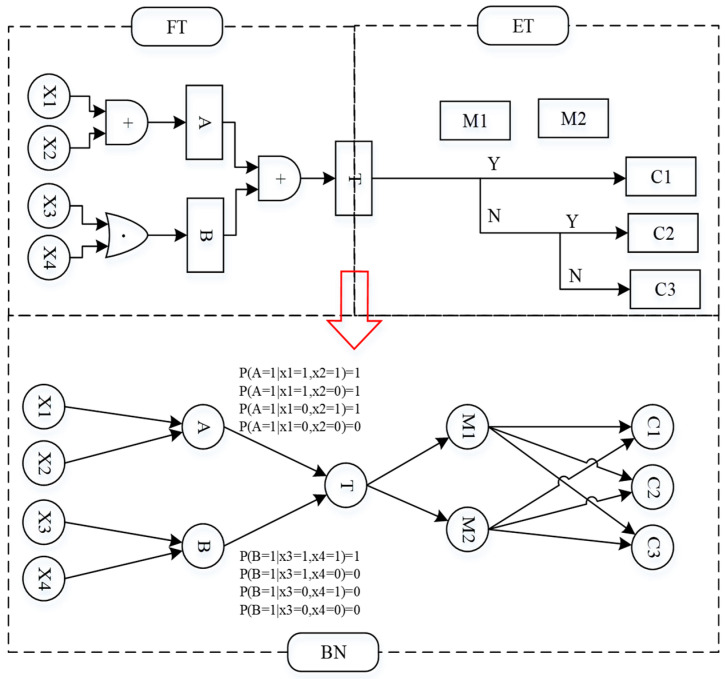
Transformation of the bow-tie model into a Bayesian network process.

**Figure 5 ijerph-19-16713-f005:**
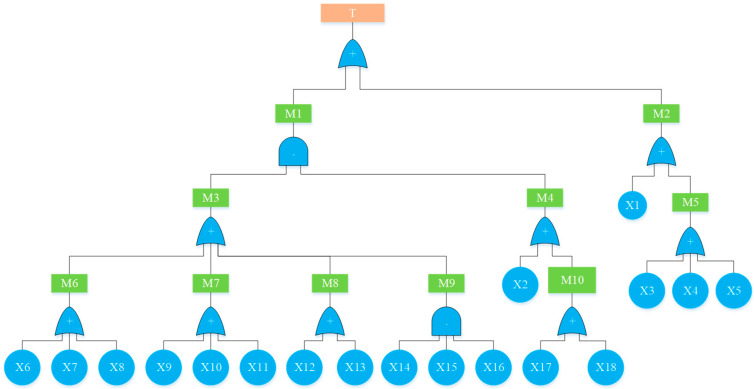
Fault tree of liquid ammonia leakage.

**Figure 6 ijerph-19-16713-f006:**
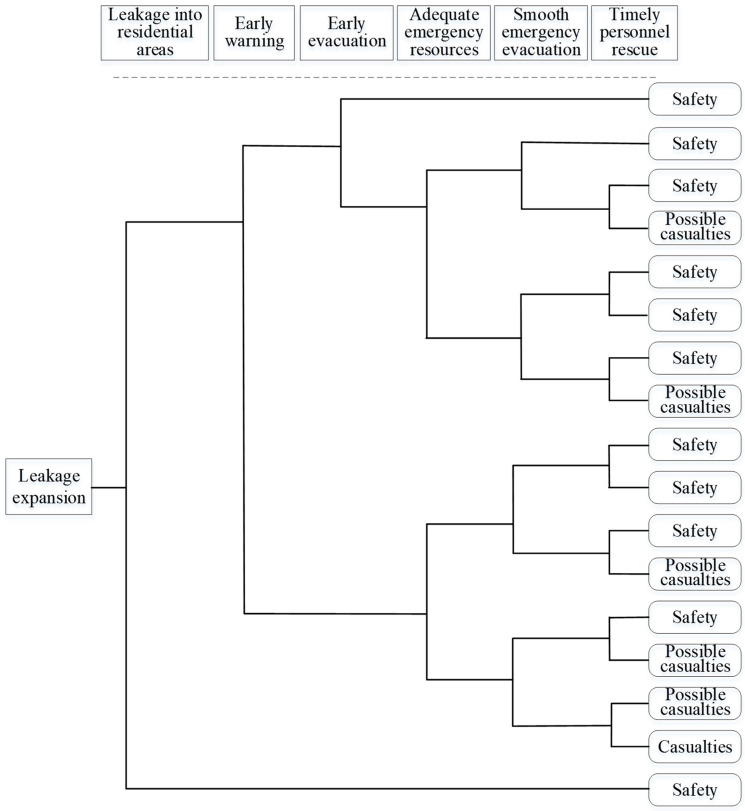
Residential area scenario event tree.

**Figure 7 ijerph-19-16713-f007:**
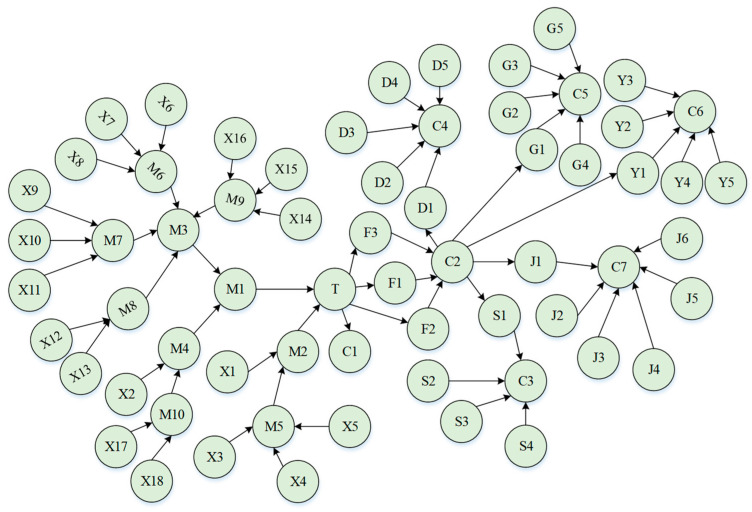
Bayesian network diagram of liquid ammonia leakage.

**Figure 8 ijerph-19-16713-f008:**
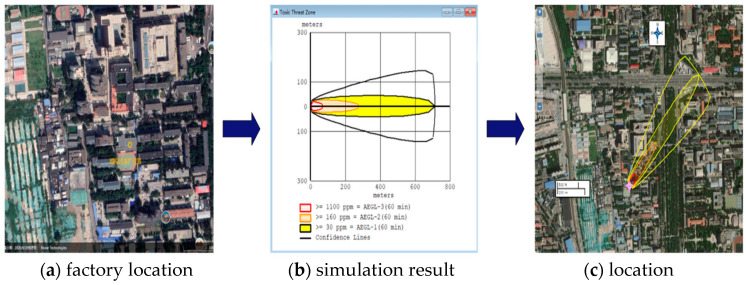
Factory location and simulation results group figure (**a**–**c**).

**Figure 9 ijerph-19-16713-f009:**
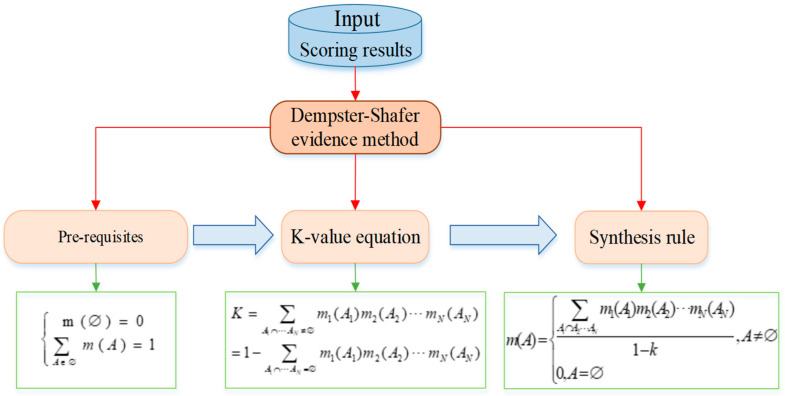
Dempster-Shafer evidence method.

**Figure 10 ijerph-19-16713-f010:**
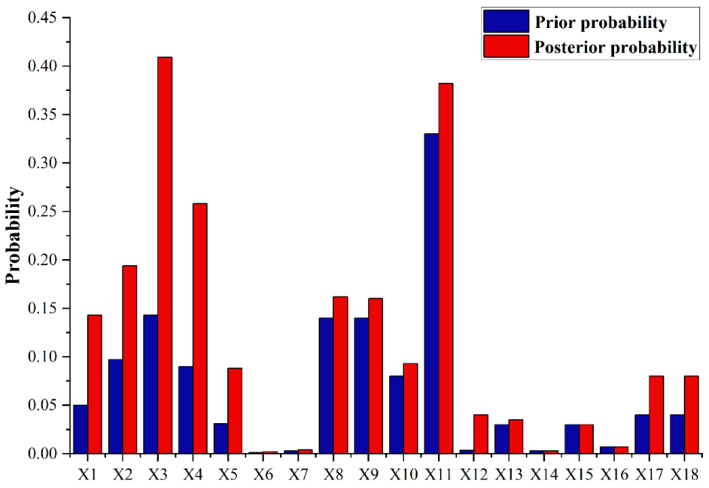
Root node probability map.

**Figure 11 ijerph-19-16713-f011:**
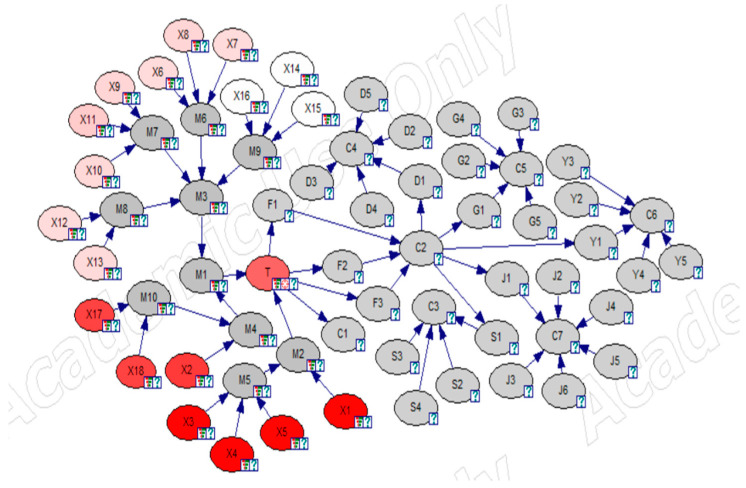
Node sensitivity analysis.

**Figure 12 ijerph-19-16713-f012:**
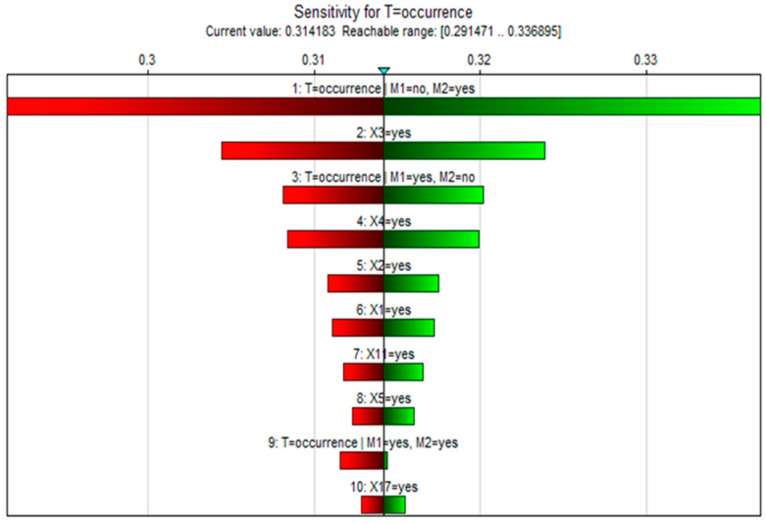
The tornado diagram dialog.

**Figure 13 ijerph-19-16713-f013:**
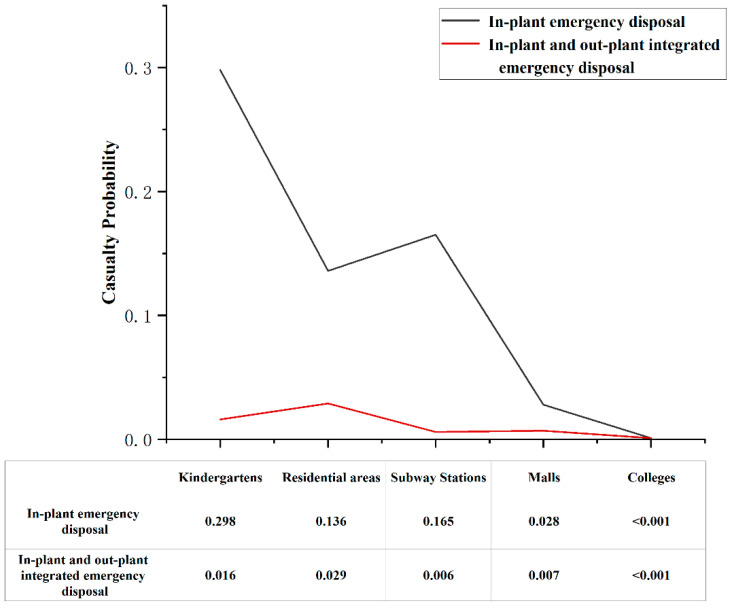
The casualty probability chart under different emergency disposal states (note: probability is 0 to 1).

**Table 1 ijerph-19-16713-t001:** Liquid ammonia leakage simulation parameters table.

Parameters	Values	Parameters	Values
Storage devices	Ammonia	Shape	Cast
Atmospheric pressure	101.3 KPa	Storage temperature	−40 °C
Storage volume	8 m^3^	Wind speed	3.5 m/s
Working pressure	1.0 MPa	Atmospheric stability	D
Ammonia ERPG1	25 ppm	Average specific heat	4.6kJ/kg·°C
Ammonia ERPG2	200 ppm	PC-TWA	20 ppm
Ammonia ERPG3	1000 ppm	PC-STEL	30 ppm
UFL	15.7%	LFL	27.4%

Note: ERPG1: causes irritation concentration; ERPG2: causes permanent damage concentration; ERPG3: lethal concentration; UFL: upper flammability limit; LFL: lower flammability limit; PC-TWA: permissible concentration-time weighted average; PC-STEL: permissible concentration-short term exposure limit.

**Table 2 ijerph-19-16713-t002:** Conditions probability of liquid ammonia leakage to colleges.

Node	Experts’ Opinion				Calculated Results
Leakage Expansion	m_1_ (1, 2)	m_2_ (1, 2)	m_3_ (1, 2)	m_4_ (1, 2)	m (1, 2)
<200 m	(0.1, 0.9)	(0.15, 0.85)	(0.15, 0.85)	(0.15, 0.85)	(0.00086, 0.99914)
200–500 m	(0.15, 0.85)	(0.2, 0.8)	(0.2, 0.8)	(0.2, 0.8)	(0.00275, 0.99725)
500–1000 m	(0.5, 0.5)	(0.45, 0.55)	(0.4, 0.6)	(0.6, 0.4)	(0.45, 0.55)
>1000 m	(0.85, 0.15)	(0.6, 0.4)	(0.5, 0.5)	(0.7, 0.3)	(0.86667, 0.13333)

Note: Probability is 0 to 1.

**Table 3 ijerph-19-16713-t003:** Partial root node derivatives and coefficient values.

Nodal Point	Derivative Value	Coefficient Value				Parameter Range
		a	b	c	d	
X3	0.6828	0.6828	0.2166	−2.22 × 10^−16^	1	[0.1286, 0.1571]
X4	0.6431	0.6431	0.2563	−2.22 × 10^−16^	1	[0.081, 0.099]
X2	0.3490	0.3491	0.2804	−2.22 × 10^−16^	1	[0.0871, 0.1064]
X1	0.6159	0.6159	0.2834	0	1	[0.0449, 0.0548]
X11	0.0740	0.0740	0.2898	0	1	[0.297, 0.363]
X5	0.6038	0.6038	0.2956	−2.22 × 10^−16^	1	[0.0277, 0.0338]
X17	0.3284	0.3284	0.3010	−4.44 × 10^−16^	1	[0.036, 0.044]

Note: a, b, c, d: the values required to calculate the derivative, calculated as D = (a × d − b × c)/(c × p + d)^2^, *p*: the numerical parameter of the node. Parameter range: displays the minimum and maximum parameter values, depending on the parameter spread.

**Table 4 ijerph-19-16713-t004:** Liquid ammonia leakage range probability.

Scope	Safety Measures Are Normal	Only Shut-Off Valve Plugging Exists	Failure of Safety Measures
<200 m	0.809	0.58	0.201
200–500 m	0.156	0.247	0.421
500–1000 m	0.034	0.142	0.286
>1000 m	<0.001	0.031	0.091

Note: Probability is 0 to 1.

**Table 5 ijerph-19-16713-t005:** Probability of casualties for each scenario within the range of different liquid ammonia leakages.

Scope	Kindergartens	Residential Areas	Subway Stations	Malls	Colleges
<200 m	0.141	0.03	0.042	0.007	<0.001
200–500 m	0.329	0.09	0.11	0.036	<0.001
500–1000 m	0.349	0.124	0.171	0.175	0.01
>1000 m	0.359	0.131	0.201	0.241	0.02

Note: Probability is 0 to 1.

**Table 6 ijerph-19-16713-t006:** Impact of emergency preparedness on changes in casualty probability across scenarios.

State	Kindergartens	Residential Areas	Subway Stations	Malls	Colleges
Normal emergency preparedness	0.36	0.13	0.208	0.25	0.023
Inadequate emergency preparedness	0.532	0.35	0.683	0.34	0.165

Note: Probability is 0 to 1.

## Data Availability

Data are available on request from the corresponding author.

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
