# Peer review of "The Scenario Construction and Evolution Method of Casualties in Liquid Ammonia Leakage Based on Bayesian Network"

_ijerph, 2022, doi:10.3390/ijerph192416713_

Round 1

Author Response

The authors would like to thank the reviewers for the constructive comments on our manuscript “The scenario construction and evolution method of casualties in liquid ammonia leakage based on Bayesian network”. The manuscript has been carefully revised following the comments.Please see the attachment.

Reviewer 2 Report

In my opinion, the Manuscript has good potential to eventually be published in the journal. That being said, there are some aspects that cause doubts to arise in my mind. I wonder a little bit what I mean. After careful consideration, I inform you that I must inform the writers of major corrections to rewrite this submission on very serious corrections grounds. Please Insert an explanation of expert opinion why this particular manuscript is not fit for publication. 

Author Response

(The authors gave the same response as above.)
